# miRNAs, from Evolutionary Junk to Possible Prognostic Markers and Therapeutic Targets in COVID-19

**DOI:** 10.3390/v14010041

**Published:** 2021-12-27

**Authors:** Brandon Bautista-Becerril, Guillermo Pérez-Dimas, Paola C. Sommerhalder-Nava, Alejandro Hanono, Julio A. Martínez-Cisneros, Bárbara Zarate-Maldonado, Evangelina Muñoz-Soria, Arnoldo Aquino-Gálvez, Manuel Castillejos-López, Armida Juárez-Cisneros, Jose S. Lopez-Gonzalez, Angel Camarena

**Affiliations:** 1Laboratorio HLA, Instituto Nacional de Enfermedades Respiratorias Ismael Cosío Villegas, Mexico City 14080, Mexico; brandon.bautistab@gmail.com (B.B.-B.); armida1109@yahoo.com.mx (A.J.-C.); 2Escuela Superior de Medicina, Departamento de Posgrado, Instituto Politécnico Nacional, Mexico City 11340, Mexico; pdimasg@gmail.com (G.P.-D.); emunozs@ipn.mx (E.M.-S.); 3Facultad de Ciencias de la Salud, Universidad Anáhuac México Norte, Mexico City 52786, Mexico; paosommerhalder@gmail.com (P.C.S.-N.); ahanono95@gmail.com (A.H.); barbie_zm@hotmail.com (B.Z.-M.); 4Organismo Público Descentralizado, Servicios de Salud Jalisco, Zapopan City 45010, Mexico; drjuliocisneros@gmail.com; 5Laboratorio de Biología Molecular, Departamento de Fibrosis Pulmonar, Instituto Nacional de Enfermedades Respiratorias Ismael Cosío Villegas, Mexico City 14080, Mexico; araquiga@yahoo.com.mx; 6Departamento de Epidemiología Hospitalaria e Infectología, Instituto Nacional de Enfermedades Respiratorias Ismael Cosío Villegas, Mexico City 14080, Mexico; mcastillejos@gmail.com; 7Laboratorio de Cáncer Pulmonar, Departamento de Enfermedades Crónico-Degenerativas, Instituto Nacional de Enfermedades Respiratorias Ismael Cosío Villegas, Mexico City 14080, Mexico; slopezgonzalez@yahoo.com

**Keywords:** miRNA, COVID-19, SARS-CoV-2, virus, prognosis, therapeutic, complications

## Abstract

The COVID-19 pandemic has been a public health issue around the world in the last few years. Currently, there is no specific antiviral treatment to fight the disease. Thus, it is essential to highlight possible prognostic predictors that could identify patients with a high risk of developing complications. Within this framework, miRNA biomolecules play a vital role in the genetic regulation of various genes, principally, those related to the pathophysiology of the disease. Here, we review the interaction of host and viral microRNAs with molecular and cellular elements that could potentiate the main pulmonary, cardiac, renal, circulatory, and neuronal complications in COVID-19 patients. miR-26a, miR-29b, miR-21, miR-372, and miR-2392, among others, have been associated with exacerbation of the inflammatory process, increasing the risk of a cytokine storm. In addition, increased expression of miR-15b, -199a, and -491 are related to the prognosis of the disease, and miR-192 and miR-323a were identified as clinical predictors of mortality in patients admitted to the intensive care unit. Finally, we address miR-29, miR-122, miR-155, and miR-200, among others, as possible therapeutic targets. However, more studies are required to confirm these findings.

## 1. Introduction

At the end of 2019, several cases of pneumonia of unknown etiology were reported in Wuhan, Hubei Province, China [1,2]. The number of cases increased quickly, and the disease spread to further regions in China, later affecting other Asian countries. It was not until January 2020 that a new coronavirus was identified and named SARS-CoV-2 by the Coronaviridae Study Group (CSG) of the International Committee on Taxonomy of Viruses, which is the causal agent of COVID-19 according to the World Health Organization (WHO) [3,4,5].

The most common symptoms reported in the literature include fever, cough, fatigue, and shortness of breath. However, symptoms such as anosmia, parosmia, myalgias, arthralgias, nausea, and diarrhea have also been reported [6]. On the other hand, approximately 5% of COVID-19 patients can deteriorate in a short period and experience other multi-organ complications that could require intensive treatment, admission to the intensive care unit (ICU), and mechanical ventilation [7,8].

Due to the possible COVID-19 complications, various biomarkers have been used clinically to monitor the evolution and prognosis of patients; however, the scales and biomarkers currently used for this pandemic are not specific for COVID-19, as they were adapted from other diseases, and it has been shown that they may underestimate the true severity of disease [9,10,11,12]. In addition, we do not have specific treatments to slow the progression of more severe forms of the disease. Thus, novel markers, such as circulating microRNAs (miRNAs), could offer new diagnostic, therapeutic, and prognostic alternatives. This review analyzes the role of miRNAs as prognostic predictors and their potential use as therapeutic targets in patients with SARS-CoV-2 infection.

## 2. Current Epidemiology

Since its appearance, the spread of the virus worldwide has been so rapid that in just three months, it spread to 114 countries and caused more than 4000 deaths, prompting the WHO to declare a pandemic on 11 March 2020 [5]. Currently, after two serious waves of infections and deaths, we find ourselves immersed in the third wave, with values that already exceed the first and threaten to reach the second, as there is no evidence of a slowdown (Figure 1) [13].

By 8 November 2021, a total of 250,533,859 confirmed cases and 5.06 million deaths were reported worldwide, with a mortality rate of 1.3 to 2.79 per 100 cases in the five countries with the highest numbers of registered deaths: the United States of America, India, Brazil, United Kingdom, and Russia, according to WHO data. Mexico has accounted for 1.5% of the total and 5.7% of the deaths worldwide, with a mortality rate of 7.5 × 100 cases [13,14]. Worldwide, differences emerged between genetic ancestry and the number of cases, hospitalizations, and deaths due to COVID-19. Table 1 shows how race and ethnicity affect these variables in different populations [15].

There are significant disparities in the rates of SARS-CoV-2 infection, COVID-19-related complications, and mortality when comparing white, Non-Hispanic patients with Natives, Asians, Blacks, or Hispanics. Overall, white, Non-Hispanic patients have a lower risk in those variables when compared to subjects with other genetic ancestries. Although a genomic vulnerability related to race could be possible, there are no available studies that suggest a genetic tendency that affects a specific race at this time in COVID-19. Sociodemographic characteristics, clinical comorbidities, and presenting features may cause this disparity [16,17,18,19]. Currently there are few studies that demonstrate that differences in gene expression are associated with race, in this context Dluzen et al., observed differences in the expression of miRNAs between hypertensive African-American women compared to hypertensive white women [20].

## 3. miRNAs

The genetic material of an organism or genome plays a central role in encoding both cell tissues and the regulatory machinery that controls cell homeostasis and internal functions [21]. Although the genome is encoded by DNA, most of it is transcribed into RNA to be able to carry out the many complex biological processes derived from the genome. However, only 1–2% of the human genome codes proteins, dividing the world of RNA into two groups: (1) RNA with coding potential and (2) RNA without coding potential, also called non-coding RNA (ncRNA). In the past, the latter was considered “evolutionary junk,” but current growing evidence suggests that it has a major impact on various molecular mechanisms, with the amount of ncRNA correlated with the complexity of an organism [22]. ncRNA is classified into small nucleolar RNA (snoRNA), miRNA, circular RNA (circRNA), and long ncRNA (lncRNA) [22,23].

The role of miRNAs is fundamental in cell development and homeostasis, acting as regulators of gene expression at the post-transcriptional level through RNA interference pathways. These gene sequences bind to the 3‘untranslated region (3′-UTR), the coding region, or the 5′-UTR of messenger RNA (mRNA), thereby inhibiting translation or facilitating its degradation [21,24]. Consequently, dysregulation of miRNA functions can lead to human disease. Recent studies have reported differentially deregulated miRNAs in various types of cancer, such as breast, lung, prostate, colon, ovarian, head and neck cancer [21]. Furthermore, it has been suggested that viral infection may affect the host’s homeostasis by regulating the expression of miRNAs, causing other genes that regulate the host’s immune response to be altered, entering a cycle of uncontrolled feedback and inflammation. This phenomenon has been investigated in multiple infectious events, such as dengue, influenza, human immunodeficiency virus 1 (HIV-1), herpes simplex virus (HSV), and hepatitis C (HCV) [25,26]. These molecules may play an important role in the diagnosis, treatment, and prognosis of SARS-CoV-2 disease.

## 4. miRNA Biogenesis

miRNAs are distributed predominantly in the intragenic and intergenic regions of the genome. They are transcribed from non-coding genes or introns of genes that code proteins. The canonical pathway for the synthesis of a mature miRNA begins in the nucleus, with RNA polymerase II synthesizing long primary transcripts (pri-miRNAs) [21,24]. pri-miRNAs are 500 to 3000 nucleotides long, characterized by a loop, a typical poly-A tail at the 3′ end, and a 7-methylguanosine cap at the 5′ end [27]. pri-miRNAs bind to members of the ribonuclease III enzyme family called double-stranded RNA specific endoribonuclease (DROSHA). Then, they attach to DiGeorge syndrome critical region 8 (DGCR8), serving as a binding protein to double-stranded RNA. Together they form the “microprocessor complex,” which cleaves pri-miRNA into a precursor miRNA (pre-miRNA) 70 to 80 nucleotides in length [21,27,28].

pre-miRNAs are actively exported from the nucleus to the cytoplasm through an exportin-5-RanGTP-dependent mechanism. In the cytoplasm, they are processed by a complex formed by Dicer protein, which is an RNase III endonuclease, and its co-factors, protein kinase R activator (PACT) or transactivation response RNA-binding protein (TRBP). This complex results in a double-stranded miRNA (ds-miRNA) 17–24 nucleotides in length, called a mature miRNA duplex. Subsequently, with the help of the Argonaute protein (AGO), one of the two strands is degraded, and the other (bound to AGO) is incorporated into the RNA-induced silencing complex (RISC). This new complex binds to complementary sequences or partial complements in the 3′- or 5′-UTR of the target mRNA. Then, the RISC recruits an adapter protein (GW182), which interacts with polyA binding proteins (PABPs), inducing the recruitment of the deadenylase complex (CCR4-NOT). The target mRNAs are destabilized by deadenylation, which leads to their degradation. It is important to remember that if the binding is partially complementary, it will lead to translational repression, whereas complete complementary binding leads to degradation of the target mRNA. Notably, miRNAs represent only about 1% of the human genome but could regulate up to 60% of all protein-coding genes and affect the transcription of several hundred mRNAs (Figure 2) [24,27,28,29].

Even though the canonical miRNA biogenesis is the classical pathway, there are several paths in miRNA creation. These include Drosha-independent and Dicer-independent pathways. In a well-recognized noncanonical route (mirtron), some debranched introns act as pre-miRNA hairpins. The mirtron pathway is Drosha/DGCR8 independent. Introns are processed to the lariat mirtrons by the spliceosome and then debranched by the lariat debranching enzyme. Afterward, they fold into pre-miRNA hairpins. Subsequently, this pre-miRNA-like form is transferred to the cytoplasm by XPO5 to continue with the canonical pathways [30].

## 5. Interaction of miRNAs with Molecular and Cellular Elements in COVID-19

The immune system is vital for the control and resolution of infections, but can also cause extensive cellular damage associated with an exaggerated immune response. There is a complex network of interactions between the virus and the host’s infected cells, in which miRNAs play a fundamental role [31].

One of the main characteristics of severe acute respiratory syndrome (SARS) is an uncontrolled systemic inflammatory response, resulting in the release of pro-inflammatory cytokines and chemokines by effector immune cells, called a cytokine storm [32]. According to findings, SARS-CoV-2 enters the body through angiotensin-converting enzyme 2 (ACE2) receptors, expressed mainly in type 2 pneumocytes. Subsequently, the virus begins to replicate and activates Toll-like receptors (TLRs), causing the production and release of multiple inflammatory cytokines, among which interleukins 1β (IL-1β) and IL-6 stand out [32,33,34]. The cytokine storm triggers an aggressive inflammatory immune response that contributes to SARS, multiple organ failure, and finally, death in severe cases of infection [34]. Patients with severe COVID-19 show increased IL-2, IL-6, IL-7, granulocyte colony-stimulating factor (G-CSF), interferon-γ-inducible protein 10 (IP-10), monocyte chemoattractant protein 1 (MCP-1), macrophage inflammatory protein 1-α (MIP-1), and tumor necrosis factor-alpha (TNFα) [35].

There is a close relationship between miRNAs and the inflammatory immune response due to their role in the regulation of cytokines, such as IL-1, IL-6, and IL-8, as well as genetic receptors that regulate endothelial function [36,37,38].

Centa et al. demonstrated that the expression of miR-26a, miR-29b, and miR-34a was significantly downregulated in the lung biopsies of patients with COVID-19 compared to individuals with other diseases. In addition, they observed an inverse correlation: diminished expression of miR-29b-3p with increased levels of IL-4 (r = −0.9130) and decreased expression of miR-26a-5p with elevated levels of IL-6 (r = −0.7360) and ICAM-1 (Intercellular cell adhesion molecule-1) (r = −0.7499). They also found that increased miR-29b-3p expression correlated with higher IL-8 levels (r = 0.7908) [39]. In addition, suppression of miR-34a in CD4+ and CD8+ T cells led to a significant increase in the cytoplasmic nuclear factor kappa-light-chain-enhancer of activated B cells (NF-κB), thus contributing to the dysregulation of interleukin production [40]. In conclusion, these findings show the relevance of miR-26a-5p, miR-29b-3p, and miR-34a in endothelial dysfunction and inflammatory processes in patients with SARS-CoV-2 [39,40].

Another study, by Tang et al., showed that downregulation of miR-146a, miR-21, and miR-142 in patients with COVID-19 promotes and aggravates the inflammatory process [41]. miR-146 negatively regulates inflammation, and its decrease causes the activation of the proinflammatory NF-κB pathway and the mitogen-activated protein (MAP) kinase pathway [42,43]. In addition, it increases the expression of interleukin-1 receptor kinase-1 (IRAK-1), IRAK-2, and tumor necrosis factor receptor-associated factor 6 (TRAF-6), which exacerbate interleukin production [44,45,46]. Similarly, recent studies have confirmed that proinflammatory cytokines (TNF/IL-12) increase miR-21 accumulation, inducing an anti-inflammatory response in macrophages. This evidence suggests that inhibition of miR-21 in leukocytes could promote the overactivation of multiple inflammatory cascades in COVID-19 [47]. Finally, decreased miR-142 induces the production of glycoprotein 130, an activator of the JAK/STAT signaling pathway, which, when bound to the IL-6 signal translator, promotes a proinflammatory state [48].

The relation between miRNAs and the production of pro-inflammatory mediators plays an essential role in the modulation of inflammation processes [49]. Salvi et al. proposed that miRNA can target cytokines directly and indirectly in autoimmune diseases; direct regulation involves targeting the cytokine mRNA, whereas indirect regulation targets proteins that influence the levels of the cytokines. If the miRNA targets a cytokine activator, the cytokine level will decrease. On the contrary, if a repressor is targeted, the cytokine level increases [50]. In this context, SARS-CoV-2 could control the immune response by depleting specific host miRNAs, improving its own viral replication cycle [51]. One mechanism proposed for the regulation of host miRNA is that SARS-CoV-2 has the ability to encode sponges that bind to host miRNA, preventing their union with the mRNA and therefore its activity. Further articles report the ability of the viral genome to sequester host miRNA and the capacity of SARS-CoV-2 proteins to regulate the expression of host miRNA. This proposed mechanism could contribute to the pro-inflammatory response and the clinical manifestations observed in COVID-19 [49].

Mitochondria display various functions in eukaryotic cells, the most important as the main source of energy in the cell. They also take part in the control of cell cycle, cellular differentiation, signal transduction, cell metabolism, apoptosis, and immune responses. Due to its role in key cellular processes, it is not surprising that viruses target mitochondria to promote their replication [52]. In the case of COVID-19, SARS-CoV-2 causes mitochondrial damage through the production of reactive oxygen species (ROS) and the stimulation of multiple inflammatory cascades [52,53]. The viral infection by SARS-CoV-2 activates miRNAs that play a fundamental role in the cytokine storm and respiratory distress syndrome. For instance, miR-2392 interacts directly with mitochondrial DNA (mtDNA), increasing the silencing of genes that code proteins such as the mitochondrial outer membrane protein (TOMM20), cytochrome c in the COX6B1 subunit of oxidase (complex IV), mitochondrial transcription factor COT-2 (NR2F2), NADH:Ubiquinone Oxidoreductase Subunit S5 (NDUFS5), COX6B1 and COX10 (complex IV assembly and structural subunits), CKMT1A (mitochondrial creatine kinase), Mitochondrial Ribosomal Protein L34 (MRPL34), adenylate kinase 4 (AK4), and methionine-R-sulfoxide reductase. These proteins usually hinder oxidative stress. However, if silenced, mitochondrial damage and the excessive release of ROS occur [53].

In addition, Yasukawa et al. identified that miR-302b and miR-372 lead to mitochondrial fragmentation since they could silence genes for dynamin-related protein 1 (DRP1) and Ras-related protein 32 (RAB32). The fragmentation leads to increased ROS production and low mitochondrial distribution that does not meet local energy demands, resulting in cell death [54,55].

Furthermore, the mitochondrial antiviral signaling protein (MAVS), which is an essential protein for innate antiviral immunity found in the outer membrane of the mitochondria, usually leads to the activation of factors of the NF pathway, interferon regulatory factor 1 (IRF1), and interferon regulatory factor 3 (IRF3). However, miR-302b and miR-372 could act as modulators of the MAVS signaling pathway by attenuating the phosphorylation of regulatory factor 3 of endogenous interferon (IRF-3). Consequently, the activation of the IFN-β and NF-κB cascades is reduced, which leads to the inhibition of type I interferon signaling pathways [54,55,56].

Moreover, miR-2392 boosts hexokinase 2 (HK2) and pyruvate kinase (PKM), both positively regulating glycolysis. HK2 produces glucose-6-phosphate and enhances guanosine diphosphate (GDP) -glucose biosynthesis. Together they support glycosylation of viral spike proteins and viral replication. The mechanism of how miR-2392 is driving these pathways is not clearly understood. Overall, the upregulated glycolysis correlates with the recently documented role of glucose metabolism in the progression of viral infection and poor outcomes in COVID-19 [53].

In addition to the host miRNAs described previously, miRNAs of viral origin (v-miRNAs) have recently been identified. Therefore, in the next section, we will review the role of v-miRNAs in the pathology of COVID-19.

## 6. miRNAs of Viral Origin in COVID-19

The interactions between miRNAs and viruses have revealed a multifaceted relationship. Viruses avoid the immune response and complete their replication cycle by taking advantage of cellular miRNAs. First, viruses interact with key proteins, such as Dicer and associated proteins, blocking the processing of miRNAs. Secondly, viruses sequester miRNAs, causing deregulation of specific target mRNAs. Next, viruses use miRNAs to redirect regulatory pathways and provide survival advantages. Finally, viruses can also directly encode miRNA precursors to target and regulate the viral replicative cycle [53]. These miRNAs of viral origin have recently been studied, and the first v-miRNAs were identified in 2004 in the Epstein—Barr virus (EBV). Since then, more than 250 v-miRNAs have been discovered. The majority are associated with DNA viruses of the herpesvirus family [57].

Computational tools and sequencing of cloned small RNA molecules are two approaches that have been used to identify v-miRNAs. So far the main functions of v-miRNAs have been attributed to immune evasion, autoregulation of the viral life cycle, and tumorigenesis; however, a better understanding of its functions is still needed [57]. Aydemir et al. identified 40 different viral miRNAs encoded from the SARS-CoV-2 genome and their targets by using bioinformatics tools. Within these targets are genes of signaling pathways, including NFKB, JAK/STAT, and TGFB; epigenetic factors JARIDs and HDACs; tumor suppressors RB1, PTEN, AKT1, and CTNNB1; transcription factors E2F1, SP1, EIF4A1, TB; CXCLproteins, interleukins, and different kinases [58]. Liu et al. also used these tools to identify potential miRNAs on the SARS-CoV-2 genome and their predicted targets. The main roles of the target genes were notch binding, single-stranded DNA endodeoxyribonuclease and deoxyribonuclease activity, cellular response to peptide hormone stimulus, and regulation of the fatty acid metabolic process. In addition, Liu et al. demonstrated the capacity of viral miRNA to target promoter regions of genes and further their expression. Afterward, they analyzed the tissue-specific enhancer sequences from the lung, intestines, spleen, liver, and heart, tissues affected by SARS-CoV-2 infection. The lung was the organ with the highest number of genes targeted by miRNA on the enhancer, followed by the spleen and gut. In addition, a total of 28 human miRNAs were predicted to be sequestered by the virus genome, modulating in this way the host biological processes. Finally, targets of virus miRNAs and host miRNAs were identified on the SARS-CoV-2 genome, suggesting the role of virus and host miRNA in the pathogenicity of SARS-CoV-2. For instance, V-miR-147 can enhance the expression of transmembrane protease enzyme serine 2 (TMPRSS2), strengthening SARS-CoV-2 infection in the gastrointestinal tract [59]

Although bioinformatics analyses predict miRNA–mRNA interaction, the biological function of this interaction needs to be verified experimentally to validate the computational findings. Co-expression of miRNA and the mRNA target need to be demonstrated. Possible methods used to corroborate this co-expression are microarray profiling, RNA-sequencing, Northern blots, and quantitative real-time PCR (RT-PCR). Then, it is necessary to investigate the direct interaction between the miRNA with a specific region in the target mRNA. A common strategy used for this is the reporter gene assay. To demonstrate how miRNAs regulate target protein expression, transfection of miRNA mimics or inhibitors into cells is performed and changes in proteins are detected through Western blot, ELISA, and immune-cytochemistry. Finally, it needs to be verified whether changes in protein expression are associated with modified biological functions and whether biochemical assays are useful for detecting changes in signaling pathways. Moreover, cell proliferation, differentiation, migration, and receptor binding can also be analyzed [60].

Meng et al. isolated v-miRNA from infected Vero E6 and Calu-3 cells with SARS-CoV-2. The v-miRNA were deep-sequenced and mapped to the SARS-CoV-2 genome. Ten v-miRNAs were identified, the most significant three derived from the N gene (v-miRNA-N-28612, v-miRNA-N-29094, and v-miRNA-N-29443). Reverse-transcription droplet digital PCR (RT-ddPCR) was performed in samples from nasopharyngeal aspirate, revealing an association between viral load and genomic RNA of SARS-CoV-2 in infected patients. Furthermore, in silico analyses were conducted to identify host target genes of the v-miRNAs [61]. A total of 15 human genes were described as potential targets. Subsequently, the transfection of small RNA mimics of validated SARS-CoV-2 v-miRNAs into Calu-3 cells and primary human peripheral blood mononuclear cells (PBMCs) was performed. The RT-qPCR analysis demonstrated the downregulation of host target transcripts ACO1, BCAS1, BNIP3L, CLDN10, DMBX1, and SNCA after the overexpression of v-miRNA-N. Additionally, binding sites of SARS-CoV-2 v-miRNA-N were found in host genes related to inflammasome activation, like IL-1, NLRP3, and caspase 1. All these studies resulted in the discovery that SARS-CoV-2 targets not only human genes involved in the inflammation process, but also genes responsible for different biological processes like apoptosis, cell cycle, signaling pathways, etc. [61].

Even though validation of v-miRNAs is challenging, it is fundamental to understand their role in physiological and pathological conditions and it is necessary for their applicability as diagnosis and prognosis biomarkers or therapeutic targets.

Next, we will describe the role of miRNAs in the progression to severe forms of SAR-CoV-2 infections and its main complications.

## 7. miRNAs as Drivers of Major Complications in COVID-19

COVID-19, in addition to causing lung disease, can also produce other extrapulmonary manifestations affecting the cardiovascular, renal, cerebrovascular, and hematological systems. A previous infection can lead to the presence of persistent symptoms after the virus has left the body [53].

The main complications of COVID-19 in the central nervous system (CNS) include acute cerebrovascular events (ACEs), encephalitis, and venous or arterial thromboembolism, in about 8 to 25% of patients with severe disease. Clinically related symptoms are headaches, seizures, behavioral disorders, and impaired consciousness [62,63,64,65]. Furthermore, up to 35% of patients have hypoxemic respiratory failure. In this context, the virus can damage the lungs in three ways: acute respiratory distress syndrome (ARDS) with diffuse alveolar damage (DAD), diffuse thrombotic alveolar microvascular occlusion, and inflammation of the airways associated with inflammatory mediators. The combination of these factors results in alveolar oxygenation, hypoxemia, and acidosis. Thus, 29 to 91% of these patients require invasive mechanical ventilation [66,67,68].

After infecting the lungs, viruses can enter the bloodstream, reach the kidneys, and damage resident cells. Therefore, patients with a severe form of the disease are more likely to suffer kidney damage. More than 29% of hospitalized patients develop acute kidney injury (AKI) with clinical signs such as proteinuria, hematuria, creatinine elevated serum, and uric nitrogen. These signs together increase the risk of hospital death up to 3.61 times. [69,70]. In addition, the virus can also reach the liver through the portal and systemic circulation. SARS-CoV-2 exerts a direct cytopathic effect on hepatocytes and cholangiocytes. The virus induces cellular stress due to low oxygen supplies or increased cytokines, as observed in sepsis. Approximately 19% of hospitalized patients with serious disease suffer from hepatic failure. Hepatic failure is linked to the activation of the coagulation and fibrinolytic pathways, a relatively low platelet count, a higher granulocyte count, and high ferritin levels [71,72].

Finally, results suggest that severe COVID-19 causes thrombocytopenia, thrombocytosis, prolonged activated partial thromboplastin time (aPTT), and increased D dimer (DD) in 60% of patients in the ICU. These findings make up disseminated intravascular coagulation (DIC) syndrome [73]. Data show that 10 to 25% of severe patients and 71.4% of non-surviving patients suffered from DIC. Clinically it presents with deep-vein thrombosis, pulmonary thromboembolism, and, to a lesser extent, hemorrhagic events [64,73,74]. Therefore, in the next section, we will address how miRNAs could drive the main complications in more serious forms of COVID-19.

### 7.1. Coagulopathy

COVID-19-associated coagulopathy is similar to other systemic coagulopathies regularly seen in severe infections, especially in DIC [64]. ElevatedDD concentrations can be found in more than 50% of COVID-19 patients and indicate a poor prognosis [75]. A study of 26 patients infected with SARS-CoV-2 compared levels of DD with the expression of different miRNAs. The upregulation of miR-424 was significant, whereas miR-103a, miR-145, and miR-885 were downregulated in patients with high DD compared to patients with low DD [76]. Tissue factor (TF) was identified as a direct target of miR-145, whereas miR-885 was found to activate the von Willebrand factor (vWF); miR-424 has been associated with hypercoagulability, whereas low levels of miR-103a are linked to deep-vein thrombosis. Although the precise mechanisms of these miRNAs have yet to be defined, miR-424 was an independent predictor of thromboembolic events in patients with COVID-19 (*p* < 0.05), whereas miR-103a independently regulated DD levels (*p* < 0.001) [76].

Coagulation abnormalities in severe COVID-19 are associated with a high risk of thrombotic vascular complications, particularly venous thromboembolism [77]. Pulmonary embolism (PE) can contribute to a sudden deterioration in oxygen exchange. Although unusual in COVID-19 patients (around 0.5%), the incidence is approximately nine times higher than in the general population [78,79]. The upregulation of miR-9 in SARS-CoV-2 infection dysregulates transcription factor EP300, which increases plasminogen activator inhibitor 1 (PAI-1). PAI-1 acts as an inhibitor of tissue plasminogen activator (tPA) and urokinase plasminogen activator (uPA) [80,81]. Usually, tPA and (uPA) are the two main plasminogen-activating serine proteases. Plasminogens are inactive precursors of a protease enzyme called plasmin that breaks down insoluble fibrin polymers at specific sites, breaking down blood clots. However, the inactivation of tPA and uPA by PAI-1 inhibits fibrinolysis and promotes a prothrombotic environment [82].

### 7.2. Lung Damage

The main complication of SARS-CoV-2 infection is ARDS, which suggests severe acute lung injury. Approximately 20% of infected patients progress to this state [32]. Evidence suggests a relationship between altered miRNA levels and lung damage in COVID-19, driven by the cytokine storm and the regulatory role that miRNAs play in the state of disease.

Data show that miR-16-2 suppresses the secretion and expression of TNF-α and IL-6 mRNA while increasing anti-inflammatory IL-10 in pulmonary macrophages under normal conditions. However, its decrease in COVID-19 patients significantly increases the secretion of pro-inflammatory factors such as IL-6, TNF-α, MCP-1, and IL-1 β, and decreases the secretion of IL-10 and transforming growth factor-β (TGF-β) [83,84]. Similarly, the importance of miR-21 at the pulmonary level has been described, since coronaviruses have binding sites for this miRNA, and SARS-CoV-2 especially could alter its circulating levels, which leads to alternative activation of macrophages, proliferation of smooth muscle cells, and remodeling of human airways through the phosphatidylinositol 3 kinase (PI3K) pathway and regulation of the PTEN protein [85,86]. Additionally, miR-21 in more seriously ill patients has been associated with more days of ventilation and vasopressors, as well as a higher rate of extracorporeal membrane oxygenation (ECMO), which could be explained by the binding of this miRNA to virus receptors or its target mRNAs, such as programmed cell death receptor 4 (PDCD4), E-cadherin, tissue inhibitor of metalloproteinase 2 (TIMP2), NF-κB, and IL-12, enhancing lung remodeling in more serious disease [85,87].

Continuing with this trend, COVID-19 patients had elevated miR-1246 levels during the acute phase; this miRNA increases the pulmonary expression of ACE2, vascular permeability, and neutrophil infiltration and enhances apoptosis, which induces lung epithelium injury [88,89]. Similarly, miR-155, present in lymphoid cells of lung tissue, was elevated in patients infected with SARS-CoV-2. This miRNA responds to many inflammatory stimuli, such as TNF-α, IL-1β, pathogen-associated molecular patterns (PAMPs), and damage-associated molecular patterns (DAMPs). miR-155 could potentiate the inflammatory response, negatively regulating IFN-α/β and degrading its mRNA by targeting the kinase 1-binding protein activated by mitogens (Mapkbp1) [83,90].

With increasing age, people are more vulnerable to viral infections, such as influenza or COVID-19. Studies have shown that in older people, the expression of miR-181a is reduced, and with it the activation of the innate and adaptive immune system [91,92,93]. It has recently been shown that miR-181a is only downregulated in severe cases of COVID-19, thus linking it to delayed viral clearance after infection similar to human immunological aging. In addition, low levels of miR-181a increase the transcription of ECA2 mRNA and TMPRSS2, which enhances viral entry. In addition, these changes promote lung injury by decreasing the amount of the Bcl-2 molecule, which regulates cell apoptosis [88,91,92,94].

Furthermore, SARS-CoV-2 has been found to downregulate the expression of miR-223 in bronchoalveolar stem cells (BASCs). Inhibition of miR-223 increases ACE2 receptors, potentiates viral replication, and activates multiple inflammatory cascades. As a result, the lung cells suffer continued destruction, increased proinflammatory cytokines, and loss of their repair capacity. These data support the study by Demiray et al., which reported significant differences in miR-223 levels between patients with mild symptoms and those in a critical state. In conclusion, lower levels of miR-223 correlate with more serious clinical manifestations of COVID-19 [95,96].

The last miRNA to consider is miR-211, related in COVID-19 to an exacerbated inflammatory response and improved affinity to target sites of viral receptors [97]. miR-211 binds to the 3′-UTR of IL-10 to inhibit its expression, contributing significantly to the cascade of pulmonary inflammation, which is why the possibility of applying specific antagomirs against miR-211 has been studied in order to attenuate the cytokine storm and decrease acute lung damage in patients with COVID-19 [98,99].

### 7.3. Cardiac Damage

SARS-CoV-2 causes cardiovascular damage through systemic inflammatory response and hypoxia. These conditions generate two types of complications: myocardial injury, similar to that caused by acute coronary syndrome and identified by increased troponin I but without evidence of electrocardiogram damage, and thrombosis. Both conditions are extremely important and have been associated with increased mortality [100].

Garg et al. evaluated serum levels of miR-21, miR-208, and miR-499, which are associated with inflammation and cardiovascular disease, observing significant increases in patients with acute COVID-19 [87]. Furthermore, miR-21 exerts an antiapoptotic effect, altering the ERK–MAP kinase signaling pathway of fibroblasts in the heart. Thus, its upregulation increases the survival of fibroblasts, causing fibrosis and hypertrophy and resulting in cardiac dysfunction [101,102]. miR-208 and miR-499 are miRNAs specific to cardiac muscle. The first regulates the expression of the heavy chain of hemoglobin β in cardiomyocytes, so its increase is associated with cardiac hypertrophy and fibrosis [103]. The second inhibits apoptosis and myocardial damage caused by hypoxia and ischemia through proteins such as p53, calcineurin, and cytosolic protein related to dynamin GTPase 1 (Drp1). It is also responsible for the structural and functional differentiation of cardiac stem cells to promote repair after damage or injury. These changes translate into constant cardiac reconstruction due to myocardial lesions caused by SARS-CoV-2 in severe infection [82]. Alteration of these miRNAs indicates a strong association between COVID-19 and the cardiovascular complications that occur in this disease [87,92,101].

The pro-inflammatory and prothrombotic state in SARS-CoV-2 infection increases the risk of ischemic cardiovascular events. Thus, an acute ST-elevation myocardial infarction (STEMI) can present as the initial clinical manifestation of COVID-19 [104,105]. Under this premise, miR-320, which is elevated in the plasma of patients infected by SARS-CoV-2, targets and inhibits the mRNA of the heat shock protein (Hsp20), thereby increasing platelet aggregation, decreasing vasorelaxation activity, and hindering contractile function due to increased apoptosis and myocardial remodeling, which could increase the risk of ischemic cardiovascular events and exacerbate heart damage if the patient has already suffered such an event [102,103,106].

Heart failure has been observed in 23% of patients with COVID-19. It is associated with a poor prognosis, since there is a higher proportion of non-survivors than survivors (52% to 12%, *p* ≤ 0.001) [105]. The regulatory pathways in which miRNAs participate could increase the risk of heart failure. Such is the case with miR-155, which increases significantly in patients with COVID-19 and is related to cardiovascular damage and significant endothelial inflammation, fibroblast proliferation, promotion of apoptosis, and cardiomyocyte pyroptosis. These changes induce hypertrophy and ventricular dysfunction, which lead to heart failure [107,108,109,110,111,112]. Additionally, miR-15b is upregulated in cases of severe COVID-19, with a higher proportion in ICU patients, and this was associated with a significantly higher incidence of arrhythmias, identifying it as a possible marker of heart damage [92,105,113,114]

ACE2 is expressed in endothelial cells, vascular smooth muscle cells, migratory angiogenic cells, cardiac fibroblasts, and many other cardiac cells. ACE2 expression is higher in severe COVID-19, as SARS-CoV-2 enters host cells through its receptor [105,115]. In this context, the miR-200 family, particularly miR-200b and miR-200c, are essential for the entry of SARS-CoV-2 into cardiovascular tissue, since its decrease in COVID-19 patients is associated with increased ACE2 expression in cardiomyocytes [116]. In addition, lower levels of miR-200b are associated with patients in severe and critical clinical conditions [96]. Finally, the expression of miR-29 in cardiac tissue is five to 12 times higher in cardiac fibroblasts than in other tissues. A reduction of this miRNA activates the TGF-β pathway, favoring the proliferation of fibroblasts that induce the expression of collagen I, collagen III, elastin (ELN), and fibrillin 1 (FBN1). These changes increase the deposition of components in the cardiac extracellular matrix, resulting in myocardial stiffness and diastolic dysfunction, which could lead to heart failure, increased damage in thrombotic events, or potentiation of new episodes [117,118]. Results indicate that the relative expression of miR-29 decreases significantly with increased severity of COVID-19. In addition, lower levels of miR-29 are correlated with a higher risk of multi-organ damage, more days of hospitalization, and a lower response to adjuvant treatments [118]. Therefore, we propose this miRNA for future investigation as a possible therapeutic target.

### 7.4. Kidney Damage

SARS-CoV-2 leads to the release of pro-inflammatory mediators, such as TGF-β, vascular endothelial growth factor (VEGF), platelet-derived growth factor (PDGF), and chemokine ligand 1 (CXCL1). These mediators cause deposition of a mesangial matrix, glomerular sclerosis, hyalinosis, and glomerular fibrosis, enhancing the pathological mechanisms for nephropathy associated with COVID-19 [119,120,121]. Many patients admitted to the hospital develop kidney complications, including glomerular nephritis and progressive chronic kidney disease, but mainly acute kidney injury (AKI), which is present in 25% of critically ill patients with SARS-CoV-2 infection, especially those with underlying comorbidities, and is associated with high mortality rates [122,123].

Elevated renal TGF-β signaling is recognized as the main driving force in renal fibrosis progression by promoting the synthesis and deposition of collagen in the extracellular matrix. TGF-β signal transduction is mediated by Smads, which transduce the extracellular stimulus to the nucleus; particularly, the activation of Smad3 is responsible for inducing the expression of fibrogenic genes such as collagens [120,124]. The decrease in miR-29 in COVID-19 is related to an increase in the degree of progression to severe forms of the disease, multiple organ damage, and more days of hospitalization [118]. This may be because TGF-β reduces the expression of miR-29, which enhances Smad3, which in turn activates cells that increase the deposition of material in the extracellular matrix, including collagen I, collagen III, fibrillin, and renal elastin, inducing remodeling and glomerular fibrosis and, on a large scale, potentiating acute kidney damage [125].

Further studies show that downregulation of miR-181a reduces renin secretion and indirectly leads to suppression of renal sympathetic nerve activity. In COVID-19, the decreased levels of miR-181a could lead to more severe episodes of hypotension in patients with sepsis or cytokine storm [41,48,99,119]. Another deregulated miRNA in SARS-CoV-2 infection is miR-18a. Its decrease furthers the production of thrombospondin-1, favoring the formation of collagen and fibrin in kidney tissue. miR-18a also reduces VEGF levels, which leads to early renal dysfunction, since the degree of fibrosis increases, and decreases the proliferation of tubular and glomerular podocytes [25,119].

In relation to age and racial groups, it was discovered that suppression of miR-145 promotes ACE2 expression in the kidney [119]. miR-145 was found to be significantly downregulated in severe COVID-19. Circulating levels were comparatively lower in the elderly and people with diabetes [57,76,106,126], which could explain why the older age group has higher rates of acute kidney damage in COVID-19.

Furthermore, the apolipoprotein L1 gene (APOL1), exclusive to humans, is expressed mainly in the kidney. This protein optimizes the function of podocytes and cell junctions, increasing the survival of glomeruli. However, APOL1 mutations (G1 and G2) have been reported to have altered function and are related to a higher incidence of kidney disease in patients of African-American or Hispanic descent [127]. Patients with G1 and G2 have upregulated miR-193a, which induces oxidative stress, which causes cell death in podocytes [128]. miRNA-193 is also elevated in ARDS, which could explain the high incidence of AKI in these racial groups [129,130]. Similarly, miR-6741 was found to target many APOL1-related genes in COVID-19, increasing pro-inflammatory cytokines IL-6, IL-10, and IL-18, and CC chemokine ligand 5 (CXCL-5). miR-6741 also promotes viral entry into the kidney and increases the risk of glomerulopathy, chronic kidney disease (CKD), AKI, and tubulointerstitial lesions [131].

Finally, studies demonstrate that miR-155 is notably upregulated in acute kidney injury [132]. This increase was found to be significantly higher in COVID-19 patients than in healthy controls and patients with influenza ARDS [87], suggesting that miR-155 could be a specific regulator of acute kidney injury in SARS-CoV-2 infection.

### 7.5. Neurological Damage

SARS-CoV-2 enters the CNS using ACE2 receptors expressed in the choroid plexus, paraventricular nuclei of the thalamus, excitatory and inhibitory neurons, and areas of the olfactory bulb. In addition, it can also enter through receptors in non-neuronal cells such as astrocytes, oligodendrocytes, and endothelial cells [133,134,135]. The direct neuronal invasion and the alteration of regulatory pathways in which the virus interferes cause the main neurological manifestations of COVID-19. The disease can cause headaches, dizziness, ataxia, and encephalopathy, and in more severe cases, meningitis/encephalitis, seizures, or even status epilepticus. Ischemic stroke and acute necrotizing encephalopathy (ANE) due to COVID-19 have also been described [136,137,138,139].

Patients with SARS-CoV-2 who experienced a concomitant stroke were mainly older and had pre-existing cardiovascular comorbidities or severe infection [140]. Kim et al. showed that elevated miR-7a, which is present in patients with COVID-19, represses the transcription of several proteins, including α-synuclein (α-Syn). This elevation contributes to neurodegenerative disease, inducing mitochondrial fragmentation, oxidative stress, and autophagy, which not only increase the risk of neuronal cell death and cerebrovascular disease (CVD) but also maximize brain damage and cognitive disability when CVD has already occurred [141]. Additional research on the mechanisms of entry of SARS-CoV-2 into nerve cells suggests that another protein/receptor, neuropilin-1 (NRP1), located mainly in the olfactory bulb, may also be involved. NRP1 is a transmembrane receptor that binds to SARS-CoV-2 proteins, improving the entry of the virus into the CNS by facilitating the fusion of membranes in neuronal tissue. In addition, it induces viral replication, neuronal death, and vWF secretion in endothelial cells, accompanied by angiogenesis and platelet activation, which exacerbates CNS damage and increases the risk of CVD [142,143,144]. Usually, miR-24 is capable of repressing the expression of NRP1; however, a decrease in this miRNA has been shown in COVID-19, which could increase the expression of NRP1 and, along with this, increase the neurological damage caused by the virus entering neuronal cells [142].

Recently it has been described that miR-124 expression in the CNS decreases as COVID-19 becomes more severe [145], thus proving it to be an essential regulator in the CNS, in addition to being a regulator of the immune system at this level and an important modulator of epileptic states by directly targeting multiple components of the TLR signaling cascade, including TLR6, myeloid differentiation factor 88 (MyD88), TRAF6, and TNF-α [126,127,146,147]. In summary, miRNAs could play a critical role in the multi-organ complications of COVID-19. In Figure 3 we summarize the most relevant interactions found in this pathology.

## 8. miRNAs as Possible Prognostic Markers in COVID-19

miRNAs display almost all the characteristics of a suitable biomarker. They are measurable with non-invasive methods, they can be detected with modern technologies and quantified with a high degree of sensitivity and specificity, and they also show a long half-life in plasma, allowing rapid and cost-effective laboratory detection. These characteristics could allow early detection of pathological states and simplify patient follow-up [148]. Therefore, research on patients with COVID-19 has shown significant differences in the expression of various miRNAs compared to healthy controls or patients with other diseases, thus identifying a direct relationship between these miRNAs and the pathophysiological alterations caused by SARS-CoV-2, demonstrating that miRNAs could be used as specific and reliable biomarkers in the prognosis of this disease.

In a study by Li et al., blood samples from 10 patients diagnosed with COVID-19 and four healthy donors were analyzed; in the first group, 35 miRNAs had higher expression and 38 lower expression. Of these, miR-16 showed a 1.6-fold increase in expression compared to controls, and miR-6501 and miR-618 showed a 1.5-fold increase. Likewise, miRNAs with lower expression were detected, among which miR-627 stood out, with a decrease in these patients of 2.3 times. Other miRNAs, such as miR-183 and miR-144, showed a 1.3-fold decrease compared to healthy controls [25]. These miRNAs are related to alterations in immune and inflammatory responses. For example, the miR-16 family is critical for T cell survival, differentiation, and proliferation, enhancing the host’s antiviral response [149]. Likewise, overexpression of miR-618 reduces the development of plasma dendritic cells from CD34+ cells, which reduces the production of inflammatory cytokines [150]. Wang et al. studied the effect of miR-183, verifying that it reduces the production of ROS and inflammatory factors IL-1β, IL-6, and TNF-α [151]. Finally, overexpression of miR-144 led to a marked increase in IL-6, TNF, and CXCL2, which demonstrates the possibility of using these miRNAs as biomarkers [152].

After establishing that SARS-CoV-2 infection alters the expression of some miRNAs, their use as possible prognostic markers continues to be investigated. Tang et al. analyzed blood samples from six patients with severe and moderate COVID-19 and from healthy donors. They identified miRNAs with increased expression, including miR-15b and miR486, only in patients with clinically severe COVID-19 [41]. Data show that miR-15b accelerates intracellular viral replication and intensifies disease severity by inhibiting CD8+ T-cell activation and repressing IL-2 and IFN-γ production [41,153,154]. Likewise, miR-486 alters the activation and differentiation of T cells, in addition to preventing the resolution of post-inflammatory edema due to higher blood vessel permeability [155,156,157,158]. In addition, miR-486 expression decreases anti-inflammatory cytokine IL-10 in CD4+ T cells [159]. In the same study by Tang et al., two miRNAs with decreased expression were identified in patients with severe COVID-19: miR-181a and miR-99a [41]. Low expression of the former increases the production of pro-inflammatory cytokines TNF-α, IL-6, IL-1β, and IL-8 [160,161], whereas the latter reduces the formation of phosphorylated protein IGF-1R, essential to the survival and proliferation of activated T cells. Both alterations weaken the host’s defense against infection by SARS-CoV-2 [162,163].

In another study, Farr et al. investigated possible differences between miRNA profiles in moderate and severe COVID-19 cases. In this analysis, the need for infected patients to receive supplemental oxygen or intubation was used as a surrogate marker for severe disease, revealing that COVID-19 patients requiring oxygenation had a significantly decreased expression of let-7e-5p, miR-766, miR-651, miR-150, and miR-4433b, suggesting that these molecules could be potential candidates to stratify patients based on severity [164].

De Gonzalo et al. compared the expression levels of circulating miRNAs among 84 hospitalized patients with patients in intensive care. miR-27a, miR-27b miR-148a, miR-199a, and miR-491 were upregulated in intensive care patients compared to ward patients. Reduced levels of miR-16, miR-92a, miR-150, miR-451a, and miR-486 were also observed in critically ill patients. They identified a signature of three miRNAs (miR-148a, miR-486, and miR-451a) linked to hospitalization in intensive care (area under the curve (95% CI) = 0.89 (0.81–0.97)) [165]. It should be noted that miR-148a and miR-486 have been associated with dysregulation of B and T lymphocytes as well as chronic inflammatory response [155,166]. In addition, a study carried out by Yang et al. confirmed that miR-451a, a suppressor of IL-6 translation, was negatively regulated in five patients with COVID-19, promoting the expression of IL-6 [167].

De Gonzalo et al. conducted an additional study to test whether the signature of circulating miRNAs constitutes a predictor of mortality in critically ill patients. They identified that miR-16, miR-92a, miR-98, miR-132, miR-192, and miR-323a were significantly decreased in patients who did not survive their stay in the intensive care unit. Of these, miR-192 and miR-323a were found to be relevant predictors of mortality during the ICU stay (area under the curve (95% CI) = 0.80 (0.64–0.96)). The discriminatory potential of this miRNA signature was higher than that observed for clinical laboratory parameters such as CRP, D-dimer, and leukocyte counts, including neutrophil and lymphocyte counts and the ratio of neutrophils to lymphocytes [165]. Note that miR-192 represses CD4+ T-cell differentiation and decreases inflammation-induced lung remodeling, and miR-323a attenuates TGF-α and TGF-β signaling and modulates fibroblast function. Therefore, when diminished, inflammation and fibroproliferation increase in the lung tissue [168,169].

In addition, McDonald et al. observed that miR-2392 is significantly associated with SARS-CoV-2 infection and that miR-2392 levels were higher for patients with more severe symptoms of the disease, this includes, intubated or those who required the ICU. Therefore, they hypothesized that miR-2392 is the main initiator of the systemic impact of the infection, which could indicate that miR-2392 is overexpressed until the virus has established itself in the host and could activate important cascades of damage related to mitochondrial death, increased pro-inflammatory factors, glycolysis, and hypoxia [53].

Lastly, Fu et al. identified and validated viral miR-nsp3-3p in a cohort of 139 COVID-19 patients and 51 healthy controls. The team investigated this viral mRNA as a possible biomarker of severe disease. miR-nsp3-3p was detected in 20 patients, in nine progressing from mild/moderate to severe disease the miR-nsp3-3p signals dropped to undetectable in the recovery stage. Results suggest that miR-nsp3-3p could predict the progression of severity 7.4 days in advance with an accuracy of 97.1% [170].

The significance of these studies lies in the possibility of forming prognostic algorithms. Human miRNAs that interact with the SARS-CoV-2 genome can be captured in currently available bioinformatics platforms to provide an analysis of the interactions between miRNAs and the prognostic markers of the disease (Figure 4).

## 9. miRNAs as Predictive Markers of Clinical Course

Additionally, miRNAs could be used as predictive markers of clinical improvement in COVID-19. In a study by Donyavi et al., differences in miRNA expression were detected between patients with acute SARS-CoV-2 infection and patients with post-acute infection. The results demonstrated that the expression of miR-29a, -146a, -155, and -let-7b in acute and post-acute samples from COVID-19 patients was significantly higher than that in healthy controls. Approximately 4–5 weeks after the acute phase of COVID-19, the expression levels of miR-29a, -146a, and let-7b were examined, which were 3.7, 2.9, and 2.7 times higher, respectively, than in the acute phase. It was concluded that these three miRNAs can help differentiate patients in the post-acute phase of COVID-19 from healthy controls, and miR-29a and miR-146a can be used to differentiate acute from post-acute COVID-19 patients. Subsequently, they correlated the increase in certain miRNAs with the expression of various signs and symptoms during acute infection. Increased miR-29a, miR-146a, and miR-155 was positively correlated with fever; miR-29a, miR-146a, miR-155, and let-7b with dry cough; and miR-146a with anosmia [171].

Additionally, Farr et al. observed that in the samples collected during the early stage of COVID-19, the virus altered the expression of 55 microRNAs encoded by the host. The results showed a strong upregulation of miR-31, -4742, and -3125 and downregulation of miR-1275, -3617, and -500b. Furthermore, the logistic regression analysis was validated, trained, and tested 1000 times to determine reproducibility and yielded a result where the measurement of miR-423-5p, miR-23a-3p, and miR-195-5p could identify COVID-19 in an early stage with 99.9% accuracy and could distinguish between COVID-19 and influenza (H1N1) with >95% accuracy, thus suggesting that the measurement of selected host molecules could predict the clinical course of the disease [164]

Finally, Zheng et al. analyzed peripheral blood samples from 18 COVID-19 patients during their treatment, convalescence, and rehabilitation. They observed that the level of expression of let-7b, miR-103a, miR-200c, and miR-2115 decreased significantly during recovery from the disease, whereas their target genes (RASGRP1, CDK6, ZEB1, and ATG5, respectively) increased, acting as important transcription factors to mediate T-cell differentiation [172].

## 10. miRNAs Associated with Response to Therapeutic Alternatives for COVID-19

The use of miRNAs can also extend to predicting the response to treatment. Sabbatinelli et al. analyzed circulating levels of miR-146a in 29 serum samples from COVID-19 patients with CT-confirmed multifocal interstitial pneumonia who required oxygen therapy and received a dose of tocilizumab. A significant increase in miR-146a levels was observed in treatment-responsive patients three days after tocilizumab administration (difference in Z-score = 1.25; *p* < 0.001), whereas no significant change was found in non-responders to treatment (*p* = 0.125) [173]. The constant downregulation of miR-146a promotes the inflammation process, as it has an anti-inflammatory effect by attenuating endothelial activation and repressing the pro-inflammatory NF-κB pathway and the MAP kinase pathway [42,43,44,45,46]. In addition, studies have shown that circulating levels of miR-146a decrease significantly with age and diabetes mellitus type 2, which could explain why the most severe cases occur in older obese patients with type 2 diabetes [174].

Likewise, Keikha et al. analyzed the plasma levels of different miRNAs from 103 patients with COVID-19 and 20 controls. Their results demonstrated that the relative expression of miR-31, miR-29a, and miR-126 decreased significantly, and the relative expression of their mRNA targets (ZMYM5, COL5A3, and CAMSAP1) increased significantly with disease severity. This pattern occurred during the hospitalization of COVID-19 patients who did not respond to remdesivir and favipiravir treatment. In patients who did respond to treatment, the expression of these miRNAs and their mRNA targets returned to normal [118]. In studies with mice, miR-31 interacted with the IL-17 receptor A 3′UTR, the IL-6 signal transducer, and the alpha mRNA of the interleukin-7 receptor subunit to suppress the early immune response [175]. In vitro, miR-29a inhibited hypoxia-induced proliferation and secretion of pulmonary adventitial fibroblasts. Additionally, it suppressed hypoxia-induced expression of α-smooth muscle actin and extracellular matrix collagen in fibroblasts of the pulmonary adventitia [176]. Finally, miR-126 inhibited the angiogenic function of human lung microvascular endothelial cells (HLMVEC) by targeting the L-type amino acid transporter (1LAT1) -mTOR signaling axis, suggesting that inhibition of miR-126 may be useful for conditions associated with microvascular loss [177].

## 11. miRNAs as Possible Therapeutic Targets

With the recent discovery of miRNAs and their key role in the pathophysiology of different diseases, the interest in exploring their properties for therapeutic purposes is expanding. SARS-CoV-2 modulates the expression of several host cell miRNAs in its favor. On the other hand, some host cell miRNAs modulate the target viral gene expression in the immune response to defend the cells [26]. According to the relationship between miRNAs and viral infection, miRNAs appear to be potential therapeutic targets in viral diseases.

Among the advantages of the therapeutic application of miRNAs are that they are small in size and that they have the ability to target hundreds of different genes, and do so in a specific manner, allowing the regulation of whole biological pathways [178,179]. In addition, miRNAs can alter the sensitivity of various drugs, so their joint administration offers a solution to drug resistance [179]. A limitation of current treatments is that not all proteins can be modulated or targeted by drugs, which is why they are known as “non-pharmacological” proteins. However, miRNAs offer the advantage of modulating the genes of these proteins, thus enabling treatment options for various diseases that at present have no cure [180].

miRNAs have been the focus of investigation in viral infections for some years. Zhou et al. provided the first evidence that miR-2911 can directly target multiple viral genes from various influenza viruses (IAVs) and thus suppress viral replication. This finding could represent a new effective therapeutic strategy to control fatal IAV infections [181]. Woods et al. demonstrated that higher expression of miR-155 in type II alveolar cells (ATII) correlated with greater severity of disease in IAV-infected mice [182]. In that sense, studies show that lipid nanoparticles could deliver miR-155-specific antagomirs to target ATII cells [183]. The deletion of miR-155 decreases bronchoalveolar lavage fluid (BALF) leukocytes and plays an anti-inflammatory role in influenza. It has also shown reduced IAV-induced lung dysfunction and injury, leading to the conclusion that using ATII cell-specific antagomiR-155 drugs could be very beneficial in reducing inflammation and destruction of lung tissue [182]. Regarding SARS-CoV-2, upregulation of miR-155 has also been observed, which could be vital for future research on miRNAs as therapeutic targets [90]. Interestingly enough, vitamin D negatively regulates miR-155, which inhibits TNFα and IL-6, and reduces the risk of SARS induced by viruses such as SARS-CoV-2 [184].

So far, no miRNA-based therapy has been approved by the FDA. However, several treatments are in the final stages of development. Among the treatments for viral infections, miravirsen and RG-101 stand out [185,186].

Miravirsen is a high-affinity antisense oligonucleotide that acts by sequestering mature miR-122, which is present in liver cells and is necessary for HCV replication. The oligonucleotides accumulate in the liver by themselves, so their transport to the liver does not require any additional conjugation [187]. A phase II study in which 36 patients received five weekly injections of miravirsen yielded promising results, reporting undetectable levels of HCV RNA. Furthermore, no dose-limiting adverse events or resistance-associated mutations were observed at the binding sites of the HCV genome with miR-122 [188]. RG-101 is an oligonucleotide inhibitor of miR-122; its uptake by the hepatocyte is favored by its binding to the asialoglycoprotein receptor. A phase 1B study reported favorable results, with the absence of detectable levels of HCV after 76 weeks of follow-up. In general, the patients did not develop severe adverse reactions [189]. Of note, two binding regions exist in the 5′-UTR of the SARS-CoV-2 genome for miR-3150b and miR-602. Therefore, it is suggested that the design of an antisense oligonucleotide similar to miravirsen could sequester these miRNAs and inhibit viral replication [190].

Alam and Lipovich contributed by developing a computational meta-analysis tool called miRCOVID-19, which identified 14 human miRNAs expressed in respiratory and immune cells with a high probability of interacting and silencing parts of the SARS-CoV-2 genome. Among them, miR-122 (whose role in HCV has already been mentioned) was identified, therefore the use of miravirsen can be considered as an alternative for the treatment of COVID-19 [191].

In therapy against COVID-19, nucleosides, nucleotides, viral nucleic acids, and enzymes/proteins involved in the replication and transcription of SARS-CoV-2 are possible treatment targets [192]. Therefore, in search of antiviral miRNAs for the treatment of COVID-19, Demirci et al., using the psRNATarget tool, identified 479 human miRNAs that target SARS-CoV2 genes, of which 67 are directed against protein S, 1 against protein E, 10 against protein M, 21 against protein N, 369 against ORF1ab, 16 against ORF3a, 13 against ORF8, 8 against ORF7a, 4 against ORF10, and 1 against ORF6; however, the antiviral function is not confirmed [193]. Unlike the previously cited work, Sardar et al. conducted a genomic study in which they only included miRNA with experimentally verified antiviral activity, identifying six miRNAs that target the SARS-CoV-2 gene: miR-let7a and miR-101, which are directed against NSP, miR-126 and miR-378 against protein N, and miR-23b and miR-98 against protein S [194].

In similar studies, Bozgeyik et al., using bioinformatics tools, identified 24 miRNAs that target the ACE2 receptor gene, and among them the miR-200 family stands out, mainly miR-200c, as a candidate for the regulation of ACE2 in respiratory cells [195]. It is important to mention that miR-200c as a therapeutic target has already been studied in animal models. Liu et al. administered antagomir directed against miR-200c and observed a significant improvement in lung injury and SARS in an induced mouse model of H5N1 virus infection [196].

Similarly, McDonald et al. sought to develop an effective treatment based on miR-antisense using the Nanoligomer platform. miR-2392 increases mitochondrial death, pro-inflammatory factors, glycolysis, hypoxia, and the risk of more serious disease. They evaluated the efficacy of treatment with miR-antisense against human miR-2392, named SBCov207, in human lung A549 cells infected with SARS-CoV-2 and showed a drastic improvement in cell viability, with an average viral inhibition of 85%. They also evaluated the toxicity of this anti-miR-2392 nanoligomer against a SARS-CoV-2 infection, showing that A549 lung cells did not show cytotoxicity up to 20 µM, that is considereda high amount. This human cell line model reaffirms that anti-miR-2392 (SBCov207) effectively inhibits SARS-CoV-2 and is not toxic at the concentrations tested. However, more studies are required to substantiate their use in more advanced clinical trials [53].

## 12. microRNA Delivery Systems

One of the main problems of miRNA-based research and therapy is the delivery to target tissues and the toxicity associated with conventional delivery vehicles [26]. Furthermore, naked miRNAs are poorly captured by cells due to their negative charge, consequently causing unwanted off-target or on-target effects, short half-life in the systemic circulation, and limited stability in the blood due to their rapid degradation or inactivation by nucleases present in the bloodstream [197]. Therefore, both viral and non-viral miRNA delivery systems are currently used, with advantages and disadvantages for each approach, which we address below [198].

Virus-based systems typically use retroviruses, lentiviruses, and adenoviruses or adeno-associated viruses (AVVs) as delivery vectors. These are modified in some specific genomic area so they cannot replicate, which makes them safer for use. The advantage of this delivery system is that it provides high transfection efficiency and a high level of constant expression of miRNAs or antagomirs [198]. When manufactured, the retroviral genes are eliminated in the vectors to create a space of up to 8 kilobases, a generous amount of genetic material [198,199]. Nonetheless, some problems such as high toxicity due to toxin production, immunogenicity that induces inflammatory response and tissue degeneration, and mutations caused by the inserted sequence restrict its uses [198]. Because of these limitations, new miRNA delivery systems such as non-viral delivery systems are being investigated.

The non-viral delivery systems use physical and chemical methods. The first group exerts external forces to make the cell membrane transiently permeable for gene delivery. These include gene guns, electroporation, hydrodynamics, ultrasound, and laser-based energy [198], whereas chemical methods employ nanocarriers. These nanocarriers are divided into three groups. In the first group are the inorganic ones, which include gold nanoparticles (AuNPs), Fe_3_O_4_-based nanoparticles, silica-based nanoparticles, and magnetic materials. The advantages they offer are their physicochemical properties, such as shape, surface area, amphiphilicity, biocompatibility, and easy surface functionalization. However, these nanosystems also have drawbacks, such as toxicity and clearance. The second group is the polymeric carriers, classified into natural and synthetic polymers. Natural polymers include polysaccharides, peptides, and proteins, which offer a better uptake, ability to mimic natural processes, and less toxicity; however, these present as a disadvantage their high immunogenicity and degradation in sera. On the other hand, synthetic polymers include polyethyleneimine (PEI), poly (lactic-co-glycolic acid)(PLGA), and dendrimers, to mention a few. These produce lower levels of damage to cell membranes and are less cytotoxic, but the transfection efficiency is low and they are poorly biodegradable. Finally, the third group is the lipid-based system, which includes cationic lipids, ionizable and helper lipids, and lipopolyplex [198,199,200,201,202,203].

In this context, lipid-based systems such as nano delivery systems have attracted the field of research due to their simple synthesis and functionalization method. They are very stable, have high loading efficiency, and have low cytotoxicity [200]. They also promote the use of miRNAs that have a wide range of therapeutic targets. A single miRNA can simultaneously regulate various mRNAs, thus facilitating the uptake of nanoparticles by receptor-mediated endocytosis and reducing the required dose and treatment side effects [197]. Lipid nanoparticles are generally easy to prepare, plus scalable procedures for large-scale manufacturing are readily available. The variations in their physicochemical characteristics are attributed to differences in chemical composition, size variability, and surface properties of the particles. With a one-step manufacturing process, large quantities of small lipid nanoparticles (<100 nm in diameter) can be produced in a few minutes [201]. Currently, these types of formulations are commercially available for miRNA delivery. Nanoparticles such as lipofectamine, SiPORT, and DharmaFECT, which are already used in vitro and in vivo, report favorable results. They offer not only biocompatibility but also minimal toxicity and effective administration [31,198].

The application of these delivery systems has been studied mainly in the treatment of cancer; however, most of the information reported corresponds to preclinical studies. The first miRNA attached to a nanocarrier to enter a phase I clinical trial was the miR-34-loaded liposome (MRX34) for the treatment of solid tumors and hematologic malignancies. Another delivery system that entered phase I clinical trials is non-living nano-sized cells known as EDV nanocells, which contain miR-16 and miR-15 for the treatment of malignant pleural mesothelioma and non-small cell lung cancer. [202].

As mentioned before, miRNAs offer enormous potential for therapeutic application; however, some of their characteristics limit their use as naked miRNAs. These limitations can be overcome by associating them with viral and non-viral delivery systems, thus improving their applicability. Although these novel systems have not currently been evaluated in COVID-19, the continuous development of new delivery systems with increased biocompatibility and specificity offers great promise. Therefore, miRNAs treated as evolutionary junk in the past could soon be the answer to numerous diseases.

## 13. Conclusions

In recent years, humanity has faced one of the deadliest pandemics in current history. Although vaccination to protect the population has become a reality, the lack of specific treatments and new SARS-CoV-2 variants have contributed to the persistence of infections and the mortality rate. It is vital to focus on research to identify the associated molecules that could modify the prognosis in COVID-19 by acting as new therapeutic targets.

miRNA molecules have been widely studied in chronic and infectious diseases such as COVID-19 because they modulate various molecules and cellular receptors that participate in the regulation of the immune response. Previous studies have shown dysregulation in expression levels of miRNAs, which are linked to the prognosis, multi-organ damage, and in some cases, mortality of COVID-19.

In this paper, we reviewed how microRNAs regulate the expression of molecules that participate in the inflammatory process in COVID-19, as well as the role they play in the complications and progression of the disease. Future research will allow some of these miRNAs to be proposed as therapeutic targets to reduce disease severity and mortality.

## Figures and Tables

**Figure 1 viruses-14-00041-f001:**
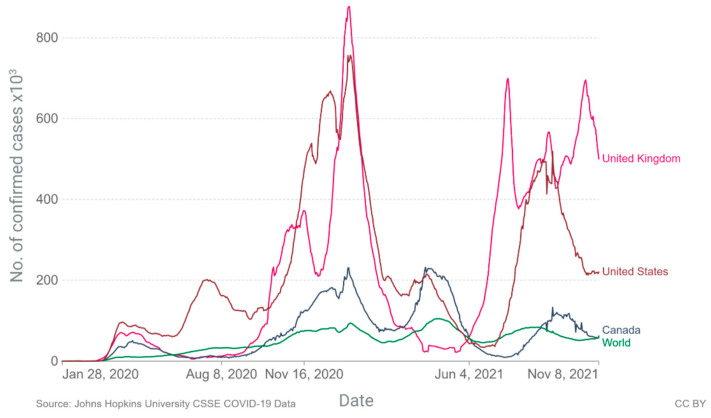
Daily new confirmed COVID-19 cases per million people from January 2020 to November 8. Created with BioRender.com.

**Figure 2 viruses-14-00041-f002:**
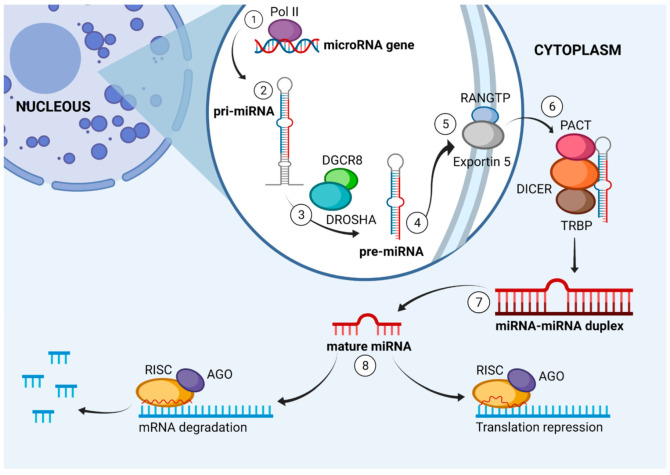
Graphical representation of the canonical miRNA biogenesis. 1. RNA polymerase II initiates long primary transcript (pri-miRNA). 2. pri-miRNA measures 500 to 3000 nucleotides in length and is shaped in the form of a loop. 3. DROSHA and DGCR8 form a microprocessor complex that forms miRNA precursor (pre-miRNA) 4. pre-miRNA is 70 to 80 nucleotides in length 5. Exportin-5 transports pre-miRNA to cytoplasm. 6. DICER and other accessory proteins convert pre-miRNA into mature miRNA 17 to 24 nucleotides in length. 8. AGO protein binds to strands of mature miRNAs and activates the RISC complex, causing repression or degradation of mRNA. Created with BioRender.com.

**Figure 3 viruses-14-00041-f003:**
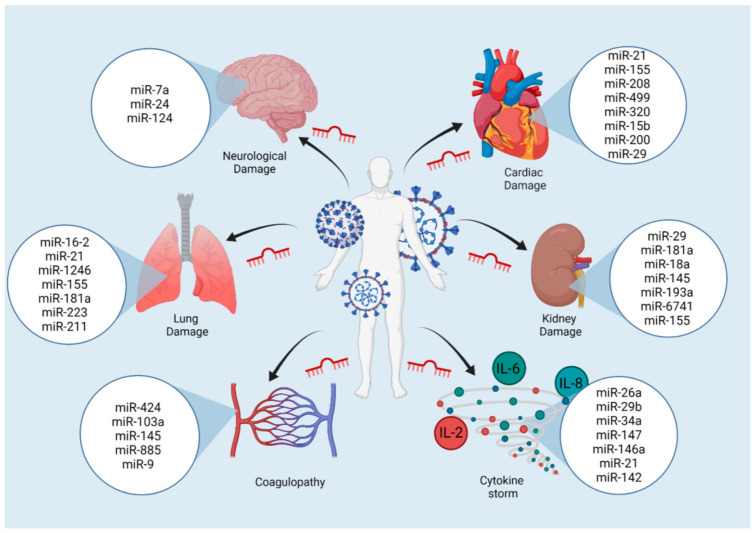
miRNAs involved in the main clinical complications of COVID-19. Created with BioRender.com.

**Figure 4 viruses-14-00041-f004:**
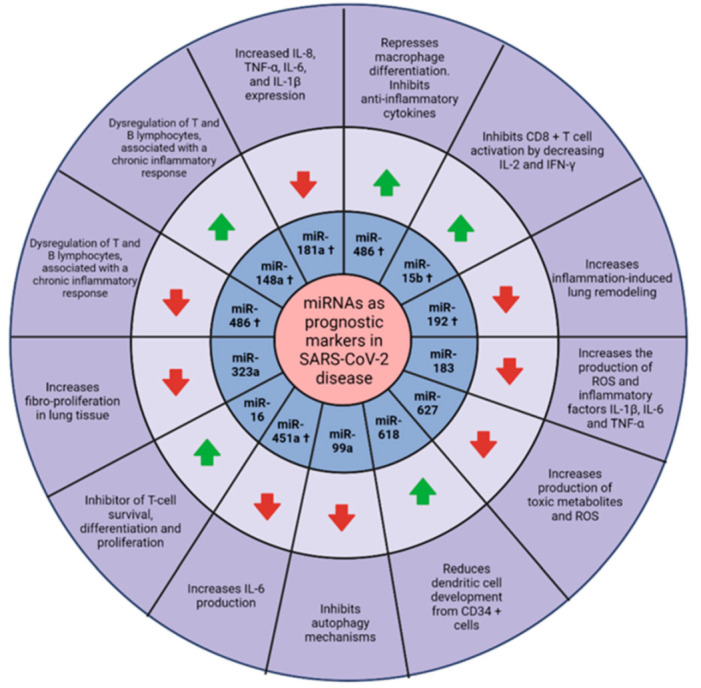
Main miRNAs reported as possible prognostic markers in COVID-19. † miRNAs specific to severe COVID-19. 
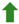
: upregulated. 
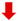
: downregulated. Created with BioRender.com.

**Table 1 viruses-14-00041-t001:** Differences with respect to ethnicity between number of cases, hospitalizations, and deaths worldwide.

Rate Ratios Compared to White, Non-Hispanic Population	American Indian or Alaskan Native	Asian	Black or African-American	Hispanic or Latino
Cases	1.7	0.7	1.1	1.9
Hospitalizations	3.4	1.0	2.8	2.8
Deaths	2.4	1.0	2.0	2.3

Differences are expressed as number of times, comparing non-Hispanic white patients and patients with different genetic ancestry.

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
