# Peer review of "miRNAs, from Evolutionary Junk to Possible Prognostic Markers and Therapeutic Targets in COVID-19"

_viruses, 2021, doi:10.3390/v14010041_

Round 1
Reviewer 1 Report
This version is ready for publication.
Reviewer 2 Report
In the article titled “miRNAs, from Evolutionary Junk to Possible Prognostic 2 Markers and Therapeutic Targets in COVID-19”, the authors have well-covered about the roles of miRNAs.
Since COVID-19 and miRNAs are a very current research area, it is important to identify new target molecules in determining appropriate treatment strategies. The study, in its current form, is acceptable in the journal Viruses.
This manuscript is a resubmission of an earlier submission. The following is a list of the peer review reports and author responses from that submission.
Round 1
Reviewer 1 Report
In the manuscript entitled, "miRNAs, from Evolutionary Junk to Possible Prognostic Markers and Therapeutic Targets in COVID-19", Bautista-Becerril and colleagues review the role of miRNAs as prognostic markers and therapeutic targets for COVID-19 disease.
The review is not exhaustive in covering all studies examining miRNA's roles in COVID-19 as a few relevant studies aren't cited and discussed, including: Farr et al. 2021, PLoS Pathogens and Tyson McDonald et al. 2021, Cell Reports.
Furthermore, the authors introduce a computational study which predicts the potential for virally encoded miRNAs, and highlight one study by Liu et al. Two other published studies should also be discussed and the results should be contrasted with the Liu study: Nur Aydemir et al. 2021, Gene Rep. and Meng et al. 2021, Cells. In addition, the authors later discuss in the review a study by Fu et al. (Cell Discovery) describing another virally encoded miRNA. To make the review more cohesive, the authors should consider grouping these studies together, and including more discussion of future studies required to verify functional roles for these putative virus derived microRNAs.
Overall, the manuscript seems a bit disjoint. It is not clear from the abstract what the review covers. Improvements in the text are required for flow and to improve the reader's comprehension. The text jumps from topic to topic without contextualizing the studies. The review might benefit from an extended introduction on COVID-19 related complications to set up the discussion of the role of miRNAs in the sequelae. In addition, there are numerous typos (a subset of them are captured in the minor points below).
While numerous studies are summarized, the authors should consider major revision of the text. In several instances, the authors should elaborate to enable the reader to follow their line of thinking. For example:
(i) the authors state: "upregulation of miR-9 in SARS-CoV-2 infection dysregulates transcription factor EP300, which increases plasminogen activator inhibitor 1 (PAI-1). PAI-1 acts as an inhibitor of tissue plasminogen activator (tPA) and urokinase plasminogen activator (uPA). As a result, inhibition of fibrinolysis promotes a prothrombotic environment." In this paragraph would be made easier to read, if the link between tPA/uPA and fibrinolysis was explained.
(ii) The authors state, "Using a computational approach, Liu et al. identified the functions of a series of viral miRNAs encoded by SARS-CoV-2. They indicated that these miRNAs could regulate the host’s immune system and inflammatory response during infection. Examples of this included the miR-147 enhancing effect of C-X-C motif chemokine ligand..." The way this text is currently written, the reader may assume that miR-147 is a virally encoded miRNA. This excerpt should be reworded to avoid confusion.
Similarly, the entire manuscript should be reworked to produce a more focused reviewed with increased clarity.
Other points include:
- For miRNA profiling studies using patient samples, it would be informative to know whether patients were unvaccinated.
- Line 35: "mir" should be "miR"
- Abstract: Authors claim there is no available treatment for COVID-19 disease. This ignores the development of monoclonal antibodies The authors should modify the abstract accordingly.
- Figure 1 - please properly format your graph so the y-axis is labeled etc.
- Data stratified by ethnicity is shown in Table 1 for cases, hospitalizations, and deaths worldwide. And a caption is provided reproducing the data in the table, which seems redundant. No conclusions are drawn from the table and no link to microRNA function is provided.
- The authors state that RNA should be grouped in two "halves". To this reader, it implies the two groups are equal in size. Perhaps the term "groups" is more appropriate.
- Line 116: "intragenic and intergenic regions" would cover the whole genome - this phrasing is awkward, and the sentence should be reworked
- The described microRNA biogenesis pathway is the canonical pathway. Other pathways for miRNA biogenesis (e.g. mirtrons) exist. The text should be modified to reflect this.
- "consumption" is not equivalent to down-regulation or degradation - please re-word
- Lines 178-184 comment on correlations between miRNA expression levels and inflammatory gene expression. Please elaborate if there are proposed mechanisms for these inverse relationships (direct targeting?), and if there is predicted direct targeting - what kind of evidence is provided
- Line 76, "Rusia" should be "Russia"
- Line 217, "MiR-424" should be "miR-424"
- Figure 3 should be labeled with the names of the specific COVID complications.
- Figure 4 - perhaps avoid using the * notation on miRNAs, as it can confuse readers as this was an older annotation to denote passenger or star strand of miRNAs.
- in the conclusion, the term "microrna genetic variants" is used; however, there is no discussion of this topic in the review
- line 574: "SARs-CoV-2" should be "SARS-CoV-2"
Reviewer 2 Report
The pathophysiology, immune reaction, and differential vulnerability of different population groups and viral host immune system evasion strategies of severe acute respiratory syndrome coronavirus 2 (SARS-CoV-2) infection are not yet well understood. In the manuscript, authors have reviewed roles of microRNAs (miRNAs) as possible prognostic markers and therapeutic targets in COVID-19. Focusing on interaction between miRNAs and molecular elements in COVID-19 in host cells is an interesting topics and the manuscript of Camarena et al. is well investigated on the topic. I recommend the manuscript is suitable for publication in the journal, and a couple of minor issues/comments will further strength their conclusion.
1). Molecular mechanisms of miRNAs in its actions are mostly lacking. I sure several groups studying on miRNAs against RNA viral infection are investigating their mechanism (PMID: 31767682 etc.). The authors should discuss how the miRNAs control host molecules against SARS-CoV-2 infection.
2). There are many papers which are reviewing the pathophysiology, immune reaction, and differential vulnerability of COVID-19 (PMIDs: 32983015, 33015593, 32982760 etc.).
3). A major challenge in applying the miRNA-based therapies is delivering these compounds to the infected site in controlled amounts without inducing toxicity. I would suggest the authors should also mention and discuss how to deliver useful candidates into our body.